# Online Neural Connectivity Estimation with Noisy Group Testing

**Anne Draelos** *
Biostatistics & Bioinformatics
Duke University
anne.draelos@duke.edu

**John M. Pearson**
Biostatistics & Bioinformatics
Electrical & Computer Engineering
Neurobiology
Duke University
john.pearson@duke.edu

## Abstract

One of the primary goals of systems neuroscience is to relate the structure of neural circuits to their function, yet patterns of connectivity are difficult to establish when recording from large populations in behaving organisms. Many previous approaches have attempted to estimate functional connectivity between neurons using statistical modeling of observational data, but these approaches rely heavily on parametric assumptions and are purely correlational. Recently, however, holographic photostimulation techniques have made it possible to precisely target selected ensembles of neurons, offering the possibility of establishing direct causal links. A naive method for inferring functional connections is to stimulate each individual neuron multiple times and observe the responses of cells in the local network, but this approach scales poorly with the number of neurons. Here, we propose a method based on noisy group testing that drastically increases the efficiency of this process in sparse networks. By stimulating small ensembles of neurons, we show that it is possible to recover binarized network connectivity with a number of tests that grows only logarithmically with population size under minimal statistical assumptions. Moreover, we prove that our approach, which reduces to an efficiently solvable convex optimization problem, can be related to Variational Bayesian inference on the binary connection weights, and we derive rigorous bounds on the posterior marginals. This allows us to extend our method to the streaming setting, where continuously updated posteriors allow for optional stopping, and we demonstrate the feasibility of inferring connectivity for networks of up to tens of thousands of neurons online.

## 1 Introduction

A long-standing problem in systems neuroscience is that of inferring the functional network structure of a population of neurons from its neural activity. That is, given a set of neural recordings, we would like to know which neurons influence which others in the system without *a priori* knowledge of their anatomical connectivity. This problem is made difficult in two ways: First, new techniques in microscopy and neural probe technology have dramatically increased the size of recorded neural populations [1, 2], posing a computational challenge. Second, the fact that typical interventions in these systems remain broad and non-specific poses problems for causal inference [3, 4].

However, recent advances in precision optics and opsin engineering have resulted in photostimulation tools capable of precisely targeting individual neurons and neuronal ensembles [5–9]. This suggests that a combination of simultaneous recording and *selective* stimulation could potentially allow for

functional dissection of large-scale neural circuits. Yet the most common methods for inferring functional connectivity are purely statistical models, applied to observational data [10, 11]. They do not consider causal inferences based on interventions (though cf. [12–14]), and often make stringent parametric assumptions, which can limit their ability to recover connectivity even in simulations [15, 16].

Here, we take a different approach to inferring functional connectivity based on targeted stimulation of small, randomly-chosen neural ensembles. We adopt the framework of group testing [17–19], an experimental design strategy that relies on simultaneous tests of multiple items. Group testing reduces the complexity of detecting rare defects (here, true connections) from linear to logarithmic in the number of units, allowing it to scale to large neural populations. We show that this approach, which makes only mild statistical assumptions, can be significantly more efficient than testing single neurons in isolation. Furthermore, we propose an efficient convex relaxation of the inference problem that is related to marginal Bayesian posteriors for the existence of individual connections. Finally, we show that an optimization scheme based on dual decomposition offers a highly parallelizable, GPU-friendly problem formulation that allows us to perform inference on a population of $10^4$ neurons in the online setting. Taken together, these ideas suggest new algorithmic possibilities for the adaptive, online dissection of large-scale neural circuits.

## 2   Network Inference as Group Testing

Our goal is to recast the problem of inferring *functional* connectivity between neurons as a group testing problem. This functional connectivity has only to do with the ability of one neuron to cause changes in the activity of another and does not imply a direct synaptic connection. Thus, two neurons may be functionally connected when no direct synaptic connection exists. In particular, we are not addressing the problem of unobserved confounders—unrecorded neurons that mediate observed interactions. Nonetheless, functional connectivity remains a quantity of intense interest, since it is likely to reflect patterns of influence and information flow in neural circuits [20, 21].

To establish conventions, it will help to consider a simple baseline protocol for establishing functional connectivity: let each test consist of stimulating a single neuron, with the test possibly repeated several times. In this setup, a stimulated neuron $i$ can be considered functionally upstream of a second neuron $j$ if $j$ typically alters its activity in response to stimulation of $i$. More precisely, we assume that there exists a test $h : \mathcal{D} \to \{0, 1\}$ that concludes from data whether stimulation of $i$ altered activity in $j$. This approach has two important advantages: First, we do not need to assume that excitation of $i$ results in excitation of $j$, only that the test detects a difference. In other words, we are not limited to excitatory connections. Second, while a given test might make parametric assumptions about the data, our subsequent analysis will be agnostic to these assumptions. Thus the ability to consider a multiplicity of tests offers us a degree of statistical flexibility not present in approaches that must rely on, e.g., linearity of synaptic contributions from different neurons. But these benefits imply a tradeoff: we will only be able to amass statistical evidence for the *existence* of such connections, and possibly their signs, but not their relative strength. We view this as a reasonable tradeoff in cases where the structure of connections is of primary concern, with the added observation that, once connections are identified, a second round of more focused testing or post-hoc methods can serve to establish strengths.

To model the effects of ensemble photostimulation, we assume that all neurons in the target set receive roughly the same light intensity, and that this intensity is sufficient to evoke a detectable response if any one of the neurons is connected to some other. Moreover, we assume that stimulation is strong enough that even, in cells receiving mixed excitatory and inhibitory connections, one will predominate. That is, given $N$ observed neurons subjected to stimulations indexed by $t$, let $\mathsf{x}_{tj} = 1$ if neuron $j$ is stimulated on round $t$, and $\mathsf{w}_{i \to j} = 1$ if neuron $i$ functionally influences neuron $j$. With these conventions, we define the predicted activation of unit $i$ as the logical OR of all the connections

$$\mathsf{a}_{ti}(\mathsf{w}) = \bigvee_{j=1}^{N} \mathsf{w}_{ij}\mathsf{x}_{tj} = \max(\mathsf{w}_{i\cdot} \odot \mathsf{x}_{t\cdot}) \tag{1}$$

and the outcome of the hypothesis test $h$ with false positive rate $\alpha$ and false negative rate $\beta$ as

$$\mathsf{y}_{ti}|(\mathsf{a}_{ti} = 1) \sim \mathrm{Bern}(1 - \beta) \qquad\qquad \mathsf{y}_{ti}|(\mathsf{a}_{ti} = 0) \sim \mathrm{Bern}(\alpha) \,. \tag{2}$$

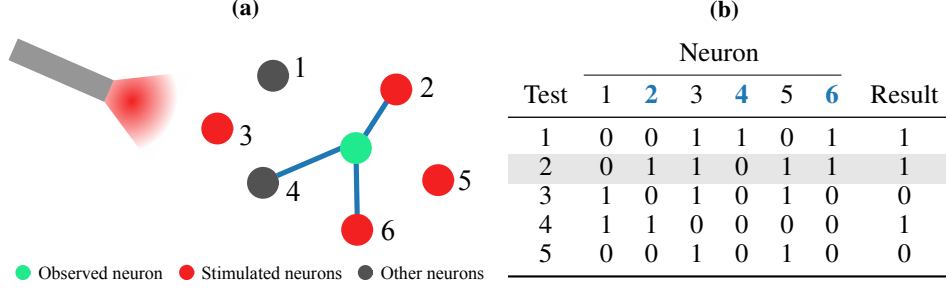

Figure 1: **Neural stimulation as group testing. (a)** Holographic photostimulation targets specific subsets of neurons (red), which result in activity in a target neuron (green). Neurons 2, 4, and 6 are functionally connected to the target neuron (blue lines), and stimulation of any one of them is sufficient to evoke activity. **(b)** Equivalent group testing matrix ($x_{ti}$), with each row a test and each column a neuron. The result of each test ($y_t$) is a logical OR of the stimulation variables for the true connections (blue). Test 2 (gray) corresponds to the stimulation in (a).

Note that this assumes a is a sufficient statistic for the outcome y, which may not hold if, e.g., false positive rates increase with the number of stimulated neurons [19].

This formulation, in which multiple units are combined into a single test that returns a positive result if any of the individual units would alone, is known as the group testing problem. Originally devised by Dorfman [17] as an efficient means of testing for syphilis in soldiers, group testing has spawned an enormous literature, with applications in medicine, communications, and manufacturing (recently reviewed in [19]). As shown by Atia and Saligrama [22], this can be cast in the language of information theory as a channel coding problem with x the codebook and y the channel output. Moreover, [22] demonstrated that when x is a randomized testing strategy to find $K$ true positives, the number of tests required to solve the problem with exponentially small average-case error is both upper and lower bounded asymptotically by $K \log N$, even when tests are noisy and $K \sim o(N)$.

The problem we consider here is more specifically one of noisy group testing in the sparse regime. That is, we allow the test to be corrupted as specified in (2) and assume $K \sim \mathcal{O}(N^{\theta})$ with $\theta \in (0, 1)$. Within this regime, approaches principally differ along two axes: adaptive versus non-adaptive test designs and the method used to infer w. In non-adaptive designs, the tests are fixed in advance, allowing them to be run in parallel at the cost of some statistical efficiency (though not necessarily asymptotically [23, 24]). Adaptive designs, by contrast, are chosen sequentially, often to optimize the information gained with each test. Below, we consider both methods, but for the remainder of this section and the next, we focus on the second axis: the method of inferring w.

For simplicity we focus on a single output neuron $j$ and its potential incoming connections $w_{ij}$ (see Fig. 1). Since the inference problems for $w_{ij}$ and $w_{ij'}$ are completely independent for $j' \neq j$, these problems can be trivially parallelized, and we drop the index $j$ in what follows. Given (1) and (2), we can infer the true connections by maximizing the total log likelihood over all $T$ tests:

$$\log p(\{y_t\}|\{w_i, x_t\}) = T \log(1 - \alpha) - \log \frac{1 - \alpha}{\alpha} \sum_t y_t \tag{3}$$

$$- \log \frac{1 - \alpha}{\beta} \sum_t a_t(w) + \log \frac{(1 - \alpha)(1 - \beta)}{\alpha\beta} \sum_t y_t a_t(w)$$

$$= \sum_t \left[ \log \frac{(1 - \alpha)(1 - \beta)}{\alpha\beta} y_t - \log \frac{1 - \alpha}{\beta} \right] a_t(w) + \text{const} ,$$

where the constant does not depend on w. For any reasonable test, we expect $1 - \beta > \alpha$ (i.e., the true positive rate exceeds the false positive rate) and $1 - \alpha > \beta$ (true negative rate exceeds false negative rate), so that the term in brackets is positive when $y_t = 1$ and negative when $y_t = 0$. Thus the maximum likelihood solution is one in which the bits $a_t(w)$ and $y_t$ most often match, similar to one-bit compressed sensing [25, 26].

Unfortunately, this integer programming problem is NP-hard in general [19], so approximate solution methods must be used. Previous approaches have used Monte Carlo methods like Gibbs Sampling

[27] and message-passing approaches like Belief Propagation [19, 28]. A third class of approaches [26] proposes to relax the binary variables $w_i \in \{0, 1\} \rightarrow w_i \in [0, 1]$ and solve a linear program to minimize $\sum_i w_i + \gamma \sum_t \xi_t$, with the $\xi_t$ slack variables representing noise (bit flips between $y_t$ and $a_t$) and $\gamma$ parameterizing the sparsity of the solution. This method indeed performs well in practice [19, 26] and makes no assumptions about the form of the noise, though it does require tuning $\gamma$, which may be difficult when the true number of defects $K$ is unknown.

Here, we propose an alternate relaxation based on independently relaxing the variables $a_t \rightarrow a_t$ and relating these to the $w_i$ via constraints. That is, instead of the $N + T$ variables $\{w_i, \xi_t\}$, we will choose to optimize over $\{w_i, a_t\}$, solving

$$\max_{\{w_i, a_t\}} \sum_t c_t a_t \quad \text{subject to} \quad x_{ti} w_i \leq a_t \leq \sum_i x_{ti} w_i, \quad w_i, a_t \in [0, 1] \tag{4}$$

with $c_t = \log \frac{(1-\alpha)(1-\beta)}{\alpha\beta} y_t - \log \frac{1-\alpha}{\beta}$. The constraints we impose on the new variables $a_t$ can be understood from (1) by noting that the maximum of a set of positive variables must be greater than or equal to each of them individually and at most equal to their sum. As we will show, this linear program in $N + T$ variables with $\sum_{ti} x_{ti} + T$ constraints may be large (and grows with the number of tests) but can nonetheless be solved efficiently even for sizable neural populations. Unfortunately, there is no guarantee that the solution to (4) produces a solution to the original integer optimization problem, and one is left with the problem of finding some method of rounding $w_i$ to produce a binary solution [29]. Fortunately, as we will argue below, this is unnecessary, and a slight alteration to (4) gives the $w_i$ an attractive interpretation.

## 3 Relaxed Group Testing as Bayesian Inference

In the discussion above we focused on maximum likelihood decoding, since this procedure has exponentially small error in the large $T$ limit [22–24, 30]. However much of this work also assumes that the number of true positives $K$ is known. In our case, by contrast, we might only have weak beliefs about the distribution of connections across neurons. Moreover, with a fixed time budget for data collection, we would benefit from the option to either stop the experiment early (if all connections have been found) or produce an estimate of uncertainty for the $w_i$ at the end of the experiment.

Thus we consider the problem of Bayesian inference for the likelihood given in (3) with Bernoulli priors $w_i \sim \text{Bern}(\pi_i)$. In this case, the log posterior takes the form

$$\log p(w|x, y) = \sum_t c_t a_t(w) + \sum_i \mu_i w_i - \log \mathcal{Z} , \tag{5}$$

with $\mu_i = \log \frac{\pi_i}{1-\pi_i}$ and $\mathcal{Z}$ a normalizing constant. Clearly, the posterior is in exponential family form, with sufficient statistics $w_i$ and $a_t(w)$. Full inference requires computation of $\mathcal{Z}$, which is practically infeasible for $N$ or $T$ large. However, we are primarily concerned with posterior (marginal) beliefs about individual connections, so we might settle for only knowing $p(w_i|x, y)$.

Luckily, two facts already mentioned allow us to compute these marginals efficiently: First, (5) is in exponential family form, and second, the $w_i$ are sufficient statistics for the posterior. Taking a Variational Bayes approach [31], we rewrite inference as an optimization problem. Let

$$q_*(w) \equiv \underset{q(w) \in \mathcal{Q}}{\arg\max} \; \mathbb{E}_q[\log p(y|w, x) + \log p(w)] + \mathcal{H}[q(w)] , \tag{6}$$

where $\mathcal{Q}$ is some class of distributions over which we optimize and $\mathcal{H} = \mathbb{E}_q[-\log q(w)]$ is the entropy. This is equivalent [31] to minimizing the KL divergence between $q_*(w)$ and $p(w|x, y)$, with $D_{KL}(q_*\|p) = 0$ if and only if $q_* = p$ almost everywhere.

We exploit the fact that we know the form of the posterior to choose a class $\mathcal{Q}$ that contains $p(w|x, y)$, since this will imply that (6) yields the true posterior. The obvious choice is to take $\mathcal{Q}$ to be the exponential family defined by the sufficient statistics $w_i$ and $a_t$. However, instead of the natural parameters corresponding to these sufficient statistics, we will define them in terms of the expectations $w_i \equiv \mathbb{E}_q[w_i]$ and $a_t \equiv \mathbb{E}_q[a_t]$. In optimization language, the latter are the primal variables and the former the duals, which are related to one another through derivatives of the free energy $\log \mathcal{Z}$ [31, 32]. With this choice, we can write

$$(\mathbb{E}[w_i], \mathbb{E}[a_t]) \equiv \underset{(w,a) \in \mathcal{M}}{\arg\max} \sum_t c_t a_t + \sum_i \mu_i w_i + \mathcal{H}(w, a) \tag{7}$$

where $\mathcal{M}$ is the marginal polytope, the set of marginals feasible under all possible distributions [32]. Clearly, since $\mathsf{w}_i$ and $\mathsf{a}_t$ are binary, we have $\mathbb{E}[\mathsf{w}_i] = P(\{\mathsf{w}_i = 1\})$, $\mathbb{E}[\mathsf{a}_t] = P(\cup_{j,\mathsf{x}_{tj}=1}\{\mathsf{w}_j = 1\})$, and the constraints in (4) follow from simple containment and union bounds for any $P$. More generally, letting $\mathcal{S}_t = \{j | \mathsf{x}_{tj} = 1\}$, there are additional consistency conditions on the $a_t$:

$$a_t \le P(\cup_{t' \in \mathcal{T}}\{\mathsf{a}_{t'} = 1\}) \le \sum_{t' \in \mathcal{T}} a_{t'} \quad \text{whenever} \quad \mathcal{S}_t \subset \bigcup_{t' \in \mathcal{T}} \mathcal{S}_{t'}. \tag{8}$$

That is, whenever any subset of trials $\mathcal{T}$ includes all neurons stimulated on trial $t$, $a_t$ is bounded above by the sum of the $a_{t'}$ from these other trials.

However, if we allow $\mathcal{H}$ to take values in $\mathbb{R} \cup \{\infty\}$, defining $\overline{\mathcal{H}}(w, a) = \infty$ for $(w, a) \notin \mathcal{M}$, then we can write

$$(\mathbb{E}[\mathsf{w}_i], \mathbb{E}[\mathsf{a}_t]) \equiv \underset{\{w_i, a_t\}}{\arg\max} \ \sum_t c_t a_t + \sum_i \mu_i w_i + \overline{\mathcal{H}}(w, a) \tag{9}$$

$$\text{s.t.} \ \mathsf{x}_{ti} w_i \le a_t \le \sum_i \mathsf{x}_{ti} w_i, \quad w_i, a_t \in [0, 1],$$

where again, $\overline{\mathcal{H}}$ incorporates the constraints in (8). This is equivalent to (4) when we assume flat priors on $\mathsf{w}_i$ ($\mu_i = 0$) and no entropy term. In other words, the relaxed $a_t$ and $w_i$ appearing in (4) are approximate *posterior probabilities* for the binary variables $\mathsf{a}_t$ and $\mathsf{w}_i$, and this relation is exact when the entropy term $\overline{\mathcal{H}}$ is included as a regularizer. Thus, solving the optimization (9) allows us to compute posterior marginals for the connections, even though we cannot write down $p(\mathsf{w}|\mathsf{x}, \mathsf{y})$.

## 4 Optimization and Online Inference

The above arguments show that posterior inference for group testing can be reduced to the variational problem (9). However, two difficulties remain: First, calculating $\overline{\mathcal{H}}(w, a)$, requires knowing the exponential family normalizing factor $\mathcal{Z}$, which is intractable in general. Second, we need an efficient method for solving (9) for very large problems. Note again that we have only been considering the case of a single output neuron, which results in a convex program with $N + T$ variables and $2N + 3T + NT$ nominal constraints (4). When generalized to the full network, we will have $N$ independent (and thus parallelizable) programs of this size, indicating both high memory and computational requirements. Yet, as we will show, further simplifications are possible that allow solutions to (9) to be implemented even for $N > 10^4$ in the online setting.

We begin by considering a slightly more general exponential family $\widetilde{\mathcal{Q}}$ in which the $\mathsf{a}_t$ as well as the $\mathsf{w}_i$ are fundamental variables, with (1) enforced by constraint:

$$\log \tilde{q}_{\eta,\nu}(\mathsf{w}, \mathsf{a}) = \sum_t \gamma_t \mathsf{a}_t + \sum_i \delta_i \mathsf{w}_i - \sum_t \eta_t \left(\mathsf{a}_t - \sum_i \mathsf{x}_{ti} \mathsf{w}_i\right) - \sum_{ti} \mathsf{x}_{ti} \nu_{ti}(\mathsf{w}_i - \mathsf{a}_t) - \log \mathcal{Z}(\eta, \nu),$$

$$\tag{10}$$

with $\nu, \eta \ge 0$. Note that this will be related to forming the Lagrangian of the problem (9), but here, we are instead defining a set of probability distributions with $\sup_{\eta,\nu \ge 0} \tilde{q} \in \mathcal{Q}' \supset \mathcal{Q}$. That is, as the constraint forces are maximized, all distributions satisfy the explicit constraints in (9), though they are not guaranteed to satisfy those in (8). We find that, in practice, this does not affect the accuracy of recovery.

What is important to note here is that the introduction of dual variables has effectively decoupled $\mathsf{w}_i$ from $\mathsf{a}_t$, since their dependency structure is a bipartite graph. Moreover, conditioned on the dual variables, the primal variables are all *independent*. Following the derivation leading to (9) we can now pose an equivalent optimization problem:

$$\sup_{\substack{\eta,\nu \ge 0 \\ w_i, a_t \in [0,1]}} \sum_t \mathcal{L}_t(a_t, \eta, \nu) + \sum_i \mathcal{L}_i(w_i, \eta, \nu), \tag{11}$$

$$\mathcal{L}_t = \left(c_t - \eta_t + \sum_i \mathsf{x}_{ti} \nu_{ti}\right) a_t + \mathcal{H}_2(a_t) \tag{12}$$

$$\mathcal{L}_i = \left(\mu_i + \sum_t \mathsf{x}_{ti} \eta_t - \sum_t \mathsf{x}_{ti} \nu_{ti}\right) w_i + \mathcal{H}_2(w_i), \tag{13}$$

with $\mathcal{H}_2(x) = -x \log x - (1-x) \log(1-x)$ the entropy of a binary variable with mean $x$ (measured in nats). The univariate maximizations over $a_t$ and $w_i$ can easily be solved numerically:

$$a_t^* = f\left(c_t - \eta_t + \sum_i \mathsf{x}_{ti} \nu_{ti}\right) \qquad w_i^* = f\left(\mu_i + \sum_t \mathsf{x}_{ti} \eta_t - \sum_t \mathsf{x}_{ti} \nu_{ti}\right), \qquad (14)$$

where $f(x) = e^x/(1+e^x)$ is the logistic function. This formulation naturally leads to a dual decomposition approach [33] in which we first maximize exactly over $w$ and $a$ then maximize (11) at the resulting optimum with respect to $\eta$ and $\nu$. Alternately, we can bound the entropy $\mathcal{H}_2$ by a quadratic (Appendix A), for which we have the solution:

$$a_t^* = \left[1 - \left(\frac{1}{2}\right)^{\sum_i \mathsf{x}_{ti}} + \frac{1}{\sigma}\left(c_t - \eta_t + \sum_i \mathsf{x}_{ti}\nu_{ti}\right)\right]_{[0,1]} \qquad (15)$$

$$w_i^* = \left[\frac{1}{2} + \frac{1}{\sigma}\left(\mu_i + \sum_t \mathsf{x}_{ti}\eta_t - \sum_t \mathsf{x}_{ti}\nu_{ti}\right)\right]_{[0,1]}, \qquad (16)$$

where $[\cdot]_{[0,1]}$ indicates truncation to the unit interval and $\sigma \in (0,4]$ is a regularization parameter. In practice, this more weakly regularized approach, which results in overconfident posteriors, performs better when binarizing $w$ to reconstruct the underlying network.

This approach is summarized in Algorithm 1. Thanks to the decoupled nature of (10), gradient updates for $\eta$ and $\nu$ can be performed in parallel, so efficient GPU implementations are possible. The key limitation for this approach is memory: while the $\nu$ matrix is sparse (effectively masked by $\mathsf{x}$), one must still maintain space for $a$, $w$, $c$, $\mu$, $\eta$, and $\nu$ for $\mathcal{O}(NST)$ parameters, with $S$ the average number of neurons stimulated per trial. Thus, while we do benefit from using first-order methods with momentum like Adam [34], these also come at the additional memory cost of $\mathcal{O}(2NST)$ running mean and variance estimates, making it impractical for systems larger than $\sim 10^3$ neurons.

---

**Algorithm 1** Dual decomposition inference

---

1: **Initialize:** $\eta_t, \nu_{ti} \leftarrow 0$
2:
3: **while** not converged **do**
4:     Solve for $a_t^*$, $w_i^*$ via (14) or (15), (16)
5:     $\eta_t \leftarrow \eta_t - \alpha(\sum_i \mathsf{x}_{ti} w_i^* - a_t^*)$
6:     $\nu_{ti} \leftarrow \nu_{ti} + \alpha \mathsf{x}_{ti}(w_i^* - a_t^*)$
7: **end while**

---

Along different lines, we can further reduce memory requirements for very large systems by simply limiting the gradient updates in Algorithm 1 to the $\eta_t$ and $\nu_{ti}$ for the most recent $\tau$ time steps. That is, for $\tau = 50$, we stop updating $\eta_2$ for $t > 52$. This halts the memory growth of the algorithm with number of tests performed, for a space complexity of $\mathcal{O}(NS\tau + 2N^2)$. As we will demonstrate in the next section, this allows us to perform inference on a network of $\sim 10^4$ neurons (one hundred million potential connections) using gradient descent with negligible loss of accuracy. In fact, our GPU implementation using CuPy [35] performed each gradient descent iteration in under 2 seconds, fast enough to perform online inference during experiments.

Finally, we note that our identification of the $w_i$ with the posterior $p(\mathsf{w}_i|\mathsf{x},\mathsf{y})$ naturally lends itself to adaptive testing. In typical adaptive algorithms, one is interested in maximizing some expected information gain or minimizing uncertainty, which can pose difficult computational problems when only point estimates are available [18, 19]. Here, however, we can trivially select those units with greatest posterior uncertainty for priority testing. In a different vein, access to calibrated uncertainties also facilitates either early stopping (when a minimum certainty is required) or optimal test allocation (when the number of tests is limited).

## 5 Experiments

We tested the performance of Algorithm 1 in both the offline (all data) and online (one test at a time) settings. In the offline setting, we considered Bernoulli designs in which each neuron was stimulated

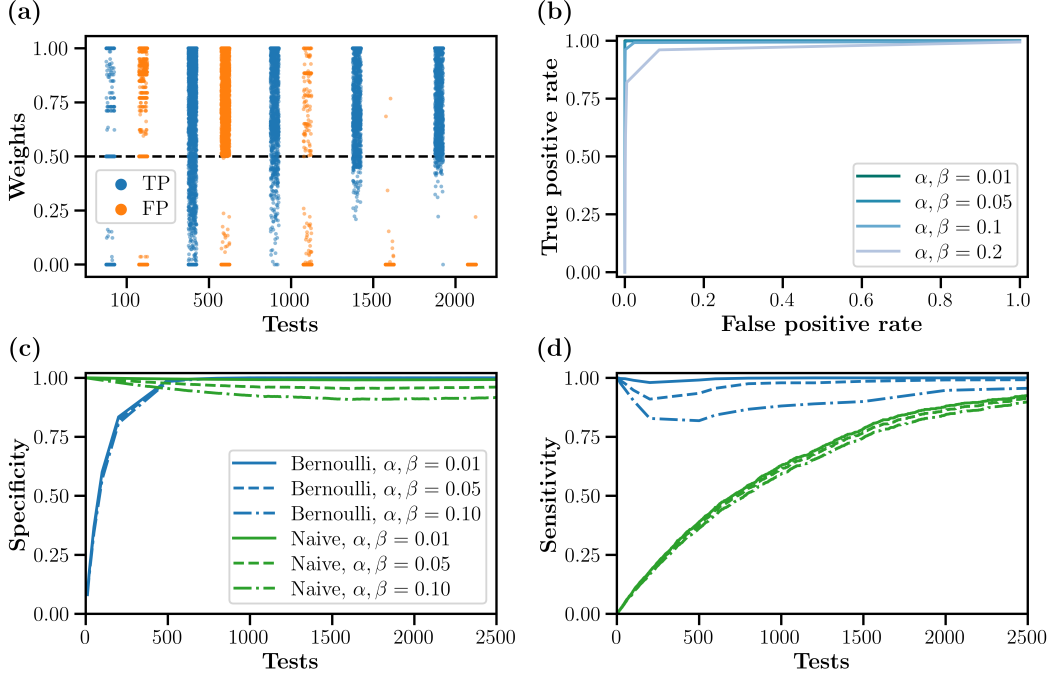

Figure 2: **Network recovery performance.** **(a)** Recovery improves with increasing numbers of tests. Dots (jittered for visibility) indicate posterior estimates for true connections (blue) and spurious connections (orange) as tests are added. The classification threshold is at 0.5 (dotted line), and we do not plot the nearly $10^6$ true negatives at 0. **(b)** ROC curves as a function of test error rates. Even as $\alpha$ and $\beta$ grow, performance degrades only moderately. **(c, d)** Specificity and sensitivity, as a function of test number and error rate. The naive approach gradually identifies positive connections, while group testing quickly separates positive and non-connections across the 0.5 threshold.

independently on each trial with probability $p_{\text{stim}} = S/N$. In the online setting, we considered both Bernoulli designs and adaptive designs, in which the top $S$ most uncertain neurons (those with $w_i$ closest to $\frac{1}{2}$) were selected for the next test. We used randomly generated binary graphs $\mathsf{w}_{ij}$ in which each link appeared independently with probability $K/N$.

We also distinguish two separate problems: uncertainty quantification and recovery. The former focuses on efficient calculation of accurate Bayesian posteriors using the formulation (9), while the latter focuses on binarizing $w$ to produce the most likely underlying w. Thus, for uncertainty we use the correct entropy $\mathcal{H}_2$ and priors defined by $\mu$, while for recovery we use the computationally cheaper quadratic approximation to $\mathcal{H}$ with $\sigma \ll 1$, $\mu = 0$ and a classification threshold at $w = \frac{1}{2}$. In our experiments, this weak regularization, which resulted in overconfident posteriors, consistently produced better recovery. The experiments presented here focus on the recovery problem. Results for uncertainty are presented in Appendix D. Unless otherwise stated, we use a **base case** of $N = 1000$, $K = N^{0.3} \approx 8$ incoming connections per neuron, $S = 10$ stimulated neurons per test, $\alpha = \beta = 0.05$, $\mu = 0$, $\sigma = 0.1$, and Adam [34] with step size 0.01, $\beta_1 = 0.9$, and $\beta_2 = 0.999$ for optimization in the offline setting, with convergence typically achieved within 50 steps.

**Sparse network recovery.** Figure 2 demonstrates the effectiveness of our algorithm in correctly recovering a binary network. The inferred system is initially regularized toward the maximum entropy solution at ($w = \frac{1}{2}$), but as the number of tests increases, connections are rapidly segregated toward 0 and 1, with classification based on a threshold at 0.5. True negatives are learned quickly at the expense of incorrectly classifying some true positives (drop in sensitivity as specificity rises), but the algorithm eventually corrects for this behavior (Fig. 2a). Tests with higher error rates show decreased performance (Fig. 2b), but this is mitigated at larger numbers of tests. Finally, in comparison with a naive model that stimulates single neurons ($S = 1$, Appendix B) group testing dominates on both measures after about 500 trials (Fig. 2c,d).

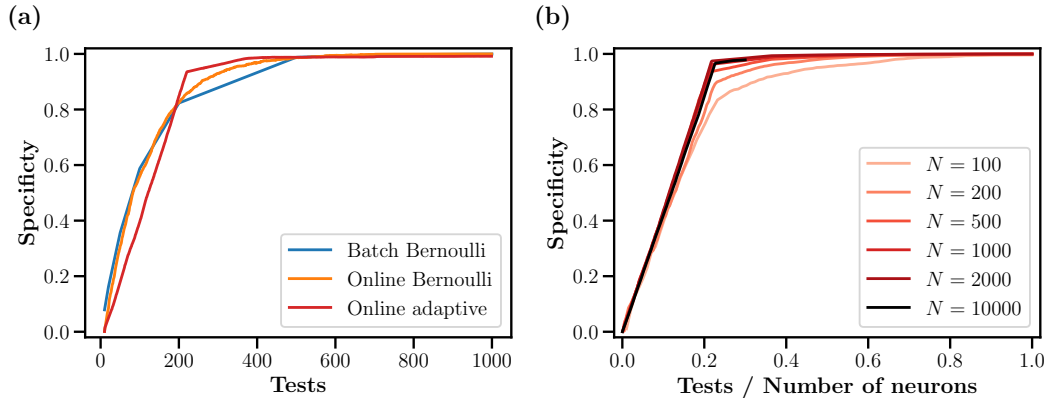

Figure 3: **Recovery in the online and adaptive settings. (a)** Specificity as a function of the number of tests for the naive, online Bernoulli, and online adaptive designs. Performance is similar to the batch case, with the online adaptive approach requiring the fewest tests overall. **(b)** Specificity as a function of the scaled number of tests $T$ (normalized $N$) for different system sizes in the adaptive case. The adaptive case exhibits an inflection point that moves toward $T \approx 0.2N$ for large $N$.

**Online performance and adaptive stimulation.** Motivated by real-time, online experimental approaches that seek to intervene in live neural circuits with photostimulation [5, 6], we also consider the online case. Here we use gradient descent (not Adam) and a sliding window of 1-10 tests to limit memory requirements and increase speed. Even with only a few fast gradient steps for each new test, we recover the network with the same level of sensitivity and specificity as in the batch case (Fig. 3). Sensitivity plots show less variation (Appendix D). This enables us to scale inference to much larger populations, even up to $N = 10^4$ (Fig. 3) with an average processing time of $< 2s$ per stimulation, for an estimated experiment time of $\sim 1.5$ hours for 2500 tests.

## 6   Discussion

We have proposed to apply noisy group testing to the problem of inferring functional connections in a neural network. We showed that a relaxation of the maximum likelihood inference problem for this setup is equivalent to Bayesian inference on the binarized network links, and that this problem can be solved efficiently for large populations in the online setting. To our knowledge, this is the first application of group testing to connectivity inference in neuroscience and the first proposal for truly scalable network inference.

Group testing itself comprises a large literature, reviewed in [18] and more recently [19]. The link between noisy group testing and information theory was established in [22–24, 30, 36] for the noise models of false positives and dilutions and in [28] for both false positives and negatives. These studies established asymptotically optimal numbers of tests maximum likelihood decoding. Linear programming relaxation as a means of efficiently solving the decoding problem was previously proposed in [26], where the objective was to identify the minimal set of positives under an arbitrary noise model. Our approach differs in relaxing both the Boolean sums $a_t$ and the defects $w_i$, as well as assuming a more specific noise model, which allows us to establish a novel connection between the solution of the relaxed convex program and Bayesian inference (4).

In neuroscience, much previous work has focused on inferring functional connectivity from correlational data, either spike trains or calcium fluorescence imaging [10, 11, 15, 37–47]. These methods typically rely on likelihood-based models and make moderate to strong parametric assumptions about the data generation process. This can result in inaccurate network recovery, even in simulation [15, 16]. Even more problematic is the difficulty of accounting for unobserved confounders [43], which can also arise in our setup when non-recorded units mediate functional connections.

Our work is similar in setup to [14], which also considered the possibilities inherent in selective stimulation of individual neurons. That work also employed a variational Bayes approach, positing a spike-and-slab prior on weights and an autoregressive generative model of ensuing calcium dynamics.

Also of note is [48], which considered optimal adaptive testing of single neurons to establish functional connections. More closely related are the approaches in [12, 49], which used a compressed sensing approach to network recovery. Those works did recover synaptic weights up to an overall normalization but did not consider either adaptive stimulation or the online inference setting. The latter problem was considered in [13], which focused on measurement of subthreshold responses in somewhat smaller systems.

By contrast with many of these approaches, ours makes relatively few statistical assumptions. We do not posit a generative or parametric model, only the existence of some statistical test for a change (not necessarily excitatory) in neuronal activity following stimulation. Moreover, our approach affords approximate Bayesian inference (which could be extended to exact inference at the cost of additional constraint forces added to (10)), does not require pretraining on existing data, and scales well to large neural populations, making it suitable for use in online settings.

However, our approach does make key assumptions that might pose challenges for experimental application. First, as Figure 2b shows, tests with poor statistical power require many more stimulations to reach correct inference, and below some threshold number of trials, this decrease in performance may be significant. Moreover, when statistical assumptions of the test $h$ are violated, the real true and false positive rates may not be known (though see Appendix C). Second, our approach ignores the relative strength of connections, as we focus on the structure of the unweighted network. This drastically reduces the number of parametric assumptions but would require a second round of more focused testing if these were quantities of interest. Nonetheless, our results suggest significant untapped potential in the application of adaptive experimental designs to large-scale neuroscience.

## Broader Impact

The focus of this work is on improving neuroscience experiments through the use of more sophisticated experimental designs. In particular, we targeted understanding the ways in which networks of neurons are constructed and function together, which has long been a focus of the field. As this advance is primarily theoretical, we do not anticipate any directly negative societal impacts. However, our work's broader impacts reflect those of neuroscience more generally: Diseases of the brain, from Alzheimer's to stroke to depression, affect a tremendous percentage of the world's population, and it is increasingly recognized that many of these conditions must be understood as pathologies of neural networks. It is our hope that circuit dissection techniques like the ones presented here will lend themselves to faster advances in our understanding of how the brain functions, with potential positive applications in the treatment of degenerative diseases. In particular, the use of implantable brain stimulation devices is now routine in the treatment of Parkinson's Disease, and it is thought that future brain-machine interfaces will help restore motor function for those suffering from paralysis. In each case, one of the key requirements is the online analysis of brain data, to which the present work represents a small contribution.

## Acknowledgments and Disclosure of Funding

Research reported in this publication was supported by a NIH BRAIN Initiative Planning Grant (R34NS116738; JP), a Ruth K. Broad Biomedical Research Foundation, Inc. Postdoctoral Fellowship Award (AD), and a Swartz Foundation Postdoctoral Fellowship for Theory in Neuroscience (AD). We especially thank Eva Naumann, who first suggested to us the idea of ensemble testing with holographic photostimulation. We would also like to thank Maxim Nikitchenko and Robert Calderbank for useful discussions and the anonymous reviewers for suggesting several improvements.

## Footnotes

*Code can be found at github.com/pearsonlab/BinaryStim

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
