[Supplementary Material]

# Supplement to: Online Connectivity Estimation with Noisy Group Testing

**Anne Draelos**
Biostatistics & Bioinformatics
Duke University
anne.draelos@duke.edu

**John M. Pearson**
Biostatistics & Bioinformatics
Electrical & Computer Engineering
Neurobiology
Duke University
john.pearson@duke.edu

## A  Entropy gradient bounds

Here, we prove the following bounds for the gradients of the entropy, the weakest (and most efficient) of which we make use of in Algorithm 1:

$$4 \left| w_i - \frac{1}{2} \right| \leq \left| \log \frac{w_i}{1 - w_i} \right| \leq |\nabla_{w_i} \mathcal{H}| \tag{1}$$

$$\leq \max \left( \left| w_i^- - \log \frac{w_i}{1 - w_i} \right|, \left| w_i^+ - \log \frac{w_i}{1 - w_i} \right| \right)$$

$$4 \left| a_t - (1 - \epsilon_t(w)) \right| \leq \left| \log \frac{a_t}{1 - a_t} - \log \left( \frac{1}{\epsilon_t(w)} - 1 \right) \right| \leq |\nabla_{a_t} \mathcal{H}| \tag{2}$$

$$\leq \max \left( \left| a_t^- - \log \frac{a_t}{1 - a_t} \right|, \left| a_t^+ - \log \frac{a_t}{1 - a_t} \right| \right) .$$

with $\epsilon_t \equiv \prod_i w_i^{\times_{ti}}$ and $w_i^\pm$, $a_t^\pm$ constants that depend on the other entries in $a$ and $w$. Note that it is these quantities, rather than the entropy itself, that are important for regularization, since overall entropy bounds may depend crucially on constants that do not affect the optimization that defines $w_i$ and $a_t$. Rather, it is the entropy gradients that define the regularization "forces" that result in estimates that are either weaker (lower bound) or stronger (upper bound) than the true entropy gradients and thus estimates of $w_i$ that are closer to or farther away from 0 and 1. Indeed, as we shall see, *both* the upper and lower bounds above derive from upper bounds on the entropy itself.

### A.1  Strong convexity bound

We start with the following Lemma:

**Lemma 1.** *For any exponential family distribution $p(\mathbf{x})$ with only Boolean sufficient statistics, $\mathcal{H}[p(\mathbf{x})]$ is $\sigma$-strongly concave for $\sigma \in (0, 4]$.*

*Proof.* Let $T_i(\mathbf{x})$ be the sufficient statistics and $\nu_i$ their natural parameters, so that

$$p(\mathbf{x}) = \frac{e^{\sum_i \nu_i T_i(\mathbf{x})}}{\mathcal{Z}} , \tag{3}$$

from which follows the well-known exponential family results

$$\frac{\partial}{\partial \nu_i} \log \mathcal{Z} = \mathbb{E}T_i \tag{4}$$

$$\frac{\partial^2}{\partial \nu_i \partial \nu_j} \log \mathcal{Z} = \frac{\partial \mathbb{E}T_i}{\partial \nu_j} = J_{ij} = \mathbb{E}[T_i T_j] - \mathbb{E}T_i \mathbb{E}T_j = \operatorname{cov}(T_i, T_j) . \tag{5}$$

That is, the Hessian of the negative free energy is both the covariance matrix of the sufficient statistics and the Jacobian of the mapping from the natural parameters to the means. Likewise, for the derivatives of the entropy,

$$\mathcal{H} = \mathbb{E}[-\log p(\mathbf{x})] = -\sum_i \nu_i \mathbb{E}T_i + \log \mathcal{Z} \tag{6}$$

$$\frac{\partial}{\partial \mathbb{E}T_j}\mathcal{H} = -\nu_j - \sum_k \frac{\partial \nu_k}{\partial \mathbb{E}T_j}\mathbb{E}T_k + \sum_k \mathbb{E}T_k \frac{\partial \nu_k}{\partial \mathbb{E}T_j} = -\nu_j \tag{7}$$

$$\frac{\partial^2}{\partial \mathbb{E}T_i \partial \mathbb{E}T_j}\mathcal{H} = -\frac{\partial \nu_j}{\partial \mathbb{E}T_i} = -J_{ij}^{-1} \ , \tag{8}$$

which is really another way of saying that $\mathcal{H}$ and $-\log \mathcal{Z}$ are convex duals, and is related to the Cramér-Rao Bound.

Now, recall that for any binary variable $T$, we have $\mathrm{var}(T) \leq \frac{1}{4}$, so the maximum eigenvalue of $\mathrm{cov}(T_i, T_j)$, which are all binary, is also $\frac{1}{4}$. From this, it follows that the minimum eigenvalue of $-\nabla^2\mathcal{H}$, which is its inverse, is at least 4.

Finally, recall that a continuously differentiable convex function $f(\mathbf{x})$ is $\sigma$-strongly convex for some $\sigma > 0$ if we have, for all $\mathbf{y}$ in $\mathrm{dom}(f)$,

$$f(\mathbf{x}) \geq f(\mathbf{y}) + \nabla f(\mathbf{y}) \cdot (\mathbf{x} - \mathbf{y}) + \frac{\sigma}{2}\|\mathbf{x} - \mathbf{y}\|^2 \ , \tag{9}$$

which is equivalent to $\nabla^2 f \succeq \sigma\mathbb{I}$ [1]. Clearly, this is true when $\sigma$ is no larger than the minimum eigenvalue of $\nabla^2 f$, and we have that $-\mathcal{H}$ is strongly convex for $\sigma \leq 4$. □

In our case, we take $\mathbf{x} = \mathsf{w}$, $T = (\mathsf{w}, \mathsf{a}(\mathsf{w}))$ and $\nu = (\nu, \gamma)$. Our plan is to expand this around the maximum of $\mathcal{H}$. This point is achieved at $\nu_i = \gamma_t = 0$ and corresponds to independent $\mathsf{w}_i$ with $w_i = \frac{1}{2}$ and $a_t = 1 - \epsilon_t(0) = 1 - \left(\frac{1}{2}\right)^{\sum_i \mathsf{x}_{ti}} \approx 1$ when the number of units tested is large. Then, from the lemma and the definition of strong convexity,

$$\mathcal{H} \leq \mathcal{H}_{sc} = N\log 2 - 2\left\|\mathbf{w} - \frac{1}{2}\right\|^2 - 2\|\mathbf{a} - 1 + \boldsymbol{\epsilon}\|^2 \ . \tag{10}$$

Finally, since we have $\mathcal{H} = \mathcal{H}_{sc}$ and $\nabla\mathcal{H} = \nabla\mathcal{H}_{sc} = \mathbf{0}$ at $w_i = a_t = 0$, and $-\nabla^2\mathcal{H} \succeq -\nabla^2\mathcal{H}_{sc}$ from above, we have $|\nabla\mathcal{H}_{sc}| \leq |\nabla\mathcal{H}|$ everywhere.

## A.2 Independent connections bound

The second, stronger lower bound can be derived by once again considering the exponential family form (3). For binary variables, we can write

$$\mathbb{E}T_i \propto e^{\nu_i}\sum_{\mathbf{x}} T_i(\mathbf{x})e^{\sum_{j\neq i}\nu_j T_j(\mathbf{x})} \propto e^{\nu_i}\mathbb{E}_{-i}T_i \ , \tag{11}$$

which gives

$$\nu_i = \log\frac{\mathbb{E}T_i}{1 - \mathbb{E}T_i} - \log\frac{\mathbb{E}_{-i}T_i}{1 - \mathbb{E}_{-i}T_i} \ . \tag{12}$$

From (7), this is $-\nabla\mathcal{H}$. The first term on the right-hand side involves expectations we assume known, while the second involves expectations in a reduced model with $\nu_i = 0$. Thus, if we were able to calculate $\mathbb{E}_{-i}T_i$, we could calculate $\nabla\mathcal{H}$ exactly. Unfortunately, this calculation is intractable in general. However, specializing to our case, if we consider $\mathcal{H}$ as a function of $(\nu, \gamma)$, then concavity gives

$$0 = |\nabla_i\mathcal{H}(0, 0)| \leq |\nabla_i\mathcal{H}(\nu, 0)| \leq |\nabla_i\mathcal{H}(\nu, \gamma)| \ . \tag{13}$$

The middle term, with $\gamma = 0$, corresponds to a model with independent $\mathsf{w}_i$, where we can easily calculate all expectations in (12), giving

$$\left|\log\frac{w_i}{1 - w_i}\right| \leq |\nabla_{w_i}\mathcal{H}| \tag{14}$$

$$\left|\log\frac{a_t}{1 - a_t} - \log\frac{1 - \prod_i w_i^{\mathsf{x}_{ti}}}{\prod_i w_i^{\mathsf{x}_{ti}}}\right| \leq |\nabla_{a_t}\mathcal{H}| \ . \tag{15}$$

Figure S1: **Comparison of entropy gradient bounds.** Plots of $\nabla\mathcal{H}$ (top) and its magnitude $|\nabla\mathcal{H}|$ (bottom) for representative (unrelated) cases of $w$ (left) and $a$ (right). Magnitude lower bounds based on strong convexity and independent w (14 - 15) are close near the maximum entropy point and diverge with distance from it. Upper (solid) and lower (dotted) bounds based on feasibility constraints (shaded region) for $(w, a)$ (17 - 20) likewise show an increasing gap near the endpoints of the interval. Importantly, the bounds for each $w_i$ depend on all $a_t$ in which it participates, and the bounds for $a_t$ depend on all $w_i$ tested. Lower bounds on $|\nabla\mathcal{H}|$ produce less regularized, optimistic estimates of $w$ and $a$, while upper bounds produce conservative estimates biased toward the maximum entropy point.

## A.3  Feasibility bounds

A final approach to bounding $|\nabla\mathcal{H}|$ again starts from (12), but this time simply bounds the second term based on mutual constraints among the parameters $w_i$ and $a_t$. That is, we again want to calculate $\mathbb{E}_{-i}T_i$, the mean of $T_i(\mathbf{x})$ under the exponential family distribution with no constraints on $T_i$ but all other sufficient statistic means specified. So, for example, we want $w_i$ calculated under the maximum entropy distribution with $(w_{i\neq j}, a_t)$ specified. Yet recall that the definition $\mathsf{a}_t \equiv \max(\mathsf{x}_{ti}\mathsf{w}_i)$ implies constraints on $a_t = \mathbb{E}\mathsf{a}_t$ and $w_i = \mathbb{E}\mathsf{w}_i$:

$$w_i\mathsf{x}_{ti} \leq a_t \leq \sum_i \mathsf{x}_{ti}w_i \ . \tag{16}$$

But this allows us to conclude that, for any $i, t$,

$$\max(\{a_t - \sum_{j\neq i}\mathsf{x}_{tj}w_j\}\cup\{0\}) \leq w_i \leq \min(\{a_t|\mathsf{x}_{ti}=1\}) \tag{17}$$

$$\max(\{\mathsf{x}_{tj}w_j\}) \leq a_t \leq \sum_i \mathsf{x}_{ti}w_i \ . \tag{18}$$

That is, if constraints dictate that $w_i \in [w_i^-, w_i^+]$, we have from (12)

$$w_i^- - \log\frac{w_i}{1-w_i} \leq \nabla\mathcal{H}_{w_i} \leq w_i^+ - \log\frac{w_i}{1-w_i} \tag{19}$$

$$\min\left(|\nabla\mathcal{H}_{w_i}^-|, |\nabla\mathcal{H}_{w_i}^+|\right) \leq |\nabla\mathcal{H}_{w_i}| \leq \max\left(|\nabla\mathcal{H}_{w_i}^-|, |\nabla\mathcal{H}_{w_i}^+|\right) \ , \tag{20}$$

with exactly analogous formulas for $a_t$. Note that the $w_i^{\pm}$ depend on *both* the other $w_j$ with which $w_i$ appears in tests *and* the $a_t$ for the tests including it, while the $a_t^{\pm}$ depend only on those connections

Figure S2: **Naive methods. (a, b)** Specificity and sensitivity, respectively, for the naive methods tested. The average case (solid green) was used in the main text.

$w_i$ tested on trial $t$. Moreover, these latter bounds allow the constraints in (8) from the main text to be included in $w^\pm$ and $a^\pm$, which can be used (at some computational cost) to derive conservative bounds on the true posteriors by upper bounding $\nabla\mathcal{H}$.

## B    Naive baseline model

As a baseline model for network recovery, we consider two versions of a naive protocol based on individual cell ($S = 1$) stimulations. For each test, a target neuron is randomly chosen (i.i.d.) from the entire population. In the first method, responses (0 or 1, according to the output of the hypothesis test) for each other neuron in the network are recorded, and these are used to update connection estimates based on a running mean. That is, the outgoing connections for the target neuron are updated each time the neuron is stimulated. All connections are initialized to zero, and connections that produce a result more than 50% of the time are set to 1. This method was used for the naive comparison in the main text.

A second analysis approach for the same stimulation protocol is to use Bayesian inference, placing Beta priors on each connection that favor non-existence (e.g., $a = 1$, $b = 10$). In this case, recovery is based on a thresholded version of the maximum a posteriori estimate given $n_1$ responses and $n_0$ non-responses to stimulation: $\mathsf{w}_{ij} = 1$ if

$$w_{\mathrm{MAP}} = \frac{a + n_1 - 1}{a + b + n_0 + n_1 - 2} > \frac{1}{2}. \tag{21}$$

If $a, b = 1$ this reduces exactly to the first naive method. The stronger the bias towards 0 in the prior, the more tests are required to correctly infer the true connections, but the number of false positives is greatly reduced.

Figure S2 shows the results of all tested naive approaches. The first method of averaging used in the main text initialized all connections to zero (solid green line); here we also show the case where all connections are initialized to 0.5 (dotted green line), and the roles of specificity and sensitivity are effectively reversed. Finally, the second method using Bayesian inference (pink) with a Beta prior ($a = 1$, $b = 5$) requires many more tests to reach the same level of sensitivity, but is most successful at remain highly specific, similar to the case of exact Bayesian inference in our new approach (see section D.1).

## C    Uncertainty in test error rates

In our model (2) in the main text, we have assumed that the true and false positive rates for our test $h$, $\alpha$ and $\beta$, are known accurately. And for many tests of interest, these two quantities may be known *theoretically*, provided the supplied data match the assumptions of the test. But when applied to real biological data assumptions are likely to be violated, and consequently, we may not know $\alpha$ and $\beta$

Figure S3: **Model misspecification. (a, b)** Specificity and sensitivity, respectively, for the base case $(\alpha, \beta = 0.05)$ in the main text (blue) and misspecified models with varying $\alpha$ and $\beta$ both over and under confident comapred to the true test error rates, including a disparate case with $\alpha = 0.0001$ and $\beta = 0.45$ (orange).

precisely. Here, we show both empirically and theoretically how this model misspecification affects our results.

## C.1 Empirical effects of misspecified error rates

Emprirically we observed essentially no difference if we misspecify the error rates; either if we assume them to be lower than they are or if we assume assume them to be higher than they are. The rates that matter are those of the test itself. Figure S3 shows a case where the assumed $\alpha$ and $\beta$ are highly disparate (0.0001 and 0.45, respectively) and the true $\alpha$ and $\beta$ are those of the base case, 0.05, as well as less disparate but still misspecified cases (e.g. $\alpha = 0.1, \beta = 0.01$). The model consistently shows a negligible difference from the fit achieved from base case, where we use the true $\alpha$ and $\beta$.

## C.2 Recovery performance is independent of error rates

From Figure S3, it is apparent that model misspecification appears to have negligible impact on network recovery. Here, we show that this is in fact the case under very mild conditions. To do this, we begin with (3) from the main text, in the $T \to \infty$ limit, so we can replace averages over trials with expectations over stimulation patterns:

$$\log p(\mathsf{y}|\mathsf{w},\mathsf{x}) = T\left[\log \frac{(1-\alpha')(1-\beta')}{\alpha'\beta'}\mathbb{E}_\mathsf{x}[\mathsf{y}\cdot\mathsf{a}(\mathsf{w},\mathsf{x})] - \log \frac{1-\alpha'}{\beta'}\mathbb{E}_\mathsf{x}[\mathsf{a}(\mathsf{w},\mathsf{x})] + \mathrm{const}\right] + o(T) \tag{22}$$

where we have *not* assumed that the error rates for the likelihood model $(\alpha', \beta')$ are the same as those for the actual data-generating process $(\alpha, \beta)$.

Fortunately, for the Bernoulli model, in which each neuron is stimulated i.i.d. with probability $p$, we can calculate the expectations in (22). Let $A = \{\mathsf{x}|\mathsf{a}(\mathsf{w},\mathsf{x}) = 1\}$, $Y = \{\mathsf{x}|\mathsf{y} = 1\}$. From the definition (1) in the main text, we have

$$\mathbb{E}_\mathsf{x}[\mathsf{a}(\mathsf{w},\mathsf{x})] = p(A) = 1 - (1-p)^\omega\,, \tag{23}$$

where $\omega = \sum_i \mathsf{w}_i$ is the number of nonzero connections. That is, the probability of predicting an activation is the complement of the probability that none of the $\omega$ connected neurons is stimulated. The second expectation can be rewritten

$$\begin{aligned}\mathbb{E}_\mathsf{x}[\mathsf{y}\cdot\mathsf{a}(\mathsf{w},\mathsf{x})] &= p(Y \cap A) \\ &= p(Y|A_*)p(A_* \cap A) + p(Y|A_*^c)p(A_*^c \cap A) \\ &= (1-\beta)\,p(A_* \cap A) + \alpha\,p(A_*^c \cap A) \\ &= (1-\beta-\alpha)\,p(A_* \cap A) + \alpha\,p(A)\,,\end{aligned} \tag{24}$$

with $*$ indicating quantities calculated in the true data-generating model, $A^c$ the complement of $A$, and in the last line, we have used $p(A^c_* \cap A) = p(A \setminus A_*) = p(A) - p(A_* \cap A)$. As for the remaining probability, we have

$$
\begin{aligned}
1 - p(A \cap A_*) = p(A^c \cup A^c_*) &= p(A^c) + p(A^c_*) - p(A^c \cap A^c_*) \\
&= (1-p)^\omega + (1-p)^{\omega_*} - (1-p)^{\omega + \omega_* - \omega_\cap}
\end{aligned}
\tag{25}
$$

where $\omega_\cap \equiv \sum_i \mathsf{w}_i \mathsf{w}_{*i} \le \min(\omega, \omega_*)$ is the number of connections shared between the true model and the model under consideration. Combining these and reinserting in (22), we then have (dropping constants)

$$
\frac{1}{T} \log p(\mathsf{y}|\mathsf{w}, \mathsf{x}) \to \mathcal{L} = c_+ (1 - \beta - \alpha)\, p(A_* \cap A) + (\alpha c_+ - c_-)\, p(A) \,,
\tag{26}
$$

where $c_\pm$ involve logarithms of $\alpha'$ and $\beta'$ as above: $c_+ = \log \frac{1-\beta'}{\alpha'} + c_-$, and $c_- = \log \frac{1-\alpha'}{\beta'}$. In other words, asymptotically, the likelihood depends on $\mathsf{w}$ only through $\omega$ and $\omega_\cap$.

Now, under a unit change in $\omega_\cap$, we have

$$
\frac{\Delta \mathcal{L}}{\Delta \omega_\cap} = c_+(1 - \beta - \alpha)(1-p)^{\omega + \omega_* - \omega_\cap}((1-p)^{-1} - 1) > 0
\tag{27}
$$

which implies that the likelihood is maximized when $\omega_\cap = \min(\omega, \omega_*)$. Substituting this into (25) gives

$$
p(A \cap A_*) = 1 - (1-p)^{\min(\omega, \omega_*)}
$$

and

$$
\mathcal{L} = \begin{cases} ((1-\beta)c_+ - c_-)\left(1 - (1-p)^\omega\right) & \omega < \omega_* \\ c_+(1 - \beta - \alpha)p(A_*) - (c_- - \alpha c_+)(1 - (1-p)^\omega) & \omega > \omega_* \end{cases} \,.
$$

which have the same optimum solution, $\omega = \omega_*$, *independently of* $c_\pm$ provided

$$
(1-\beta)c_+ > c_- > \alpha c_+
\tag{28}
$$

or

$$
\frac{1-\alpha}{\alpha} > \frac{\log \frac{1-\beta'}{\alpha'}}{\log \frac{1-\alpha'}{\beta'}} > \frac{\beta}{1-\beta} \,.
\tag{29}
$$

Of course, if $\alpha$, $\beta < 0.5$ and $\alpha' = \beta'$, this is *always* satisfied. In this case, likelihood maximization remains consistent even for a misspecified model, and we do not need accurate estimates of our test error rates to recover the true set of connections.

### C.3   Bayesian analysis of uncertain error rates

In a full Bayesian analysis, we can consider placing priors on the test error rates:

$$
\alpha \sim \text{Beta}(\phi_+, \phi_-)
\tag{30}
$$
$$
\beta \sim \text{Beta}(\varphi_+, \varphi_-)
\tag{31}
$$

Combining this with (2), we again have (3) from the main text, but we must now marginalize over our uncertainty in $\alpha$ and $\beta$. That is, we want

$$
\begin{aligned}
p(\mathsf{y}|\mathsf{w}, \mathsf{x}) &= \int p(\mathsf{y}|\mathsf{w}, \mathsf{x}, \alpha, \beta)p(\alpha)p(\beta)\, d\alpha\, d\beta \\
&= \frac{B(\phi_+ + n_{\text{FP}}, \phi_- + n_{\text{TN}})B(\varphi_+ + n_{\text{FN}}, \varphi_- + n_{\text{TP}})}{B(\phi_+, \phi_-)B(\varphi_+, \varphi_-)} \,,
\end{aligned}
\tag{32}
$$

where $B(x, y)$ is the beta function, $n_{\text{TP}}$ is the number of true positives ($\mathsf{a}_t = 1$, $\mathsf{y}_t = 1$), and similarly for the other expressions. We would like to relate this quantity to (3). The easiest way to do this is to consider the limit of large numbers of tests, so that the beta functions are given by Stirling's approximation to $\Gamma(x)$. That is,

$$
B(x, y) \sim \sqrt{2\pi}\, \frac{x^{x - \frac{1}{2}} y^{y - \frac{1}{2}}}{(x+y)^{x + y - \frac{1}{2}}} \,,
\tag{33}
$$

so that (32) gives

$$\log p(\mathsf{y}|\mathsf{w},\mathsf{x}) = n_{\text{FP}} \log\left(\frac{\phi_+ + n_{\text{FP}}}{\phi_+ + \phi_- + n_{\text{FP}} + n_{\text{TN}}}\right) + n_{\text{TN}} \log\left(\frac{\phi_- + n_{\text{TN}}}{\phi_+ + \phi_- + n_{\text{FP}} + n_{\text{TN}}}\right)$$
$$+ n_{\text{FN}} \log\left(\frac{\varphi_+ + n_{\text{FN}}}{\varphi_+ + \varphi_- + n_{\text{FN}} + n_{\text{TP}}}\right) + n_{\text{TP}} \log\left(\frac{\varphi_- + n_{\text{FN}}}{\varphi_+ + \varphi_- + n_{\text{FN}} + n_{\text{TP}}}\right)$$
$$- \frac{1}{2}\log(\phi_+ + \phi_- + n_{\text{FP}} + n_{\text{TN}}) - \frac{1}{2}\log(\varphi_+ + \varphi_- + n_{\text{FN}} + n_{\text{TP}}) + \text{constant}\,, \quad (34)$$

which can be put into correspondence with (3) (up to subleading logarithmic terms in $n$) if we identify

$$\bar{\alpha} = \frac{\phi_+ + n_{\text{FP}}}{\phi_+ + \phi_- + n_{\text{FP}} + n_{\text{TN}}} \tag{35}$$

$$\bar{\beta} = \frac{\varphi_+ + n_{\text{FN}}}{\varphi_+ + \varphi_- + n_{\text{FN}} + n_{\text{TP}}}\,. \tag{36}$$

Of course (35) and (36) are just the posterior means of $\alpha$ and $\beta$, and we see that in the limit of large numbers of tests, the logarithmic terms in $n$ can be ignored relative to the linear terms and the log evidence concentrates around the parameters of the data generating process. This in turn suggests an empirical Bayes approach in which we alternate variational inference (with $n$s fixed) with adjustment of the $n$s based on posterior estimates of the $\mathbb{E}\mathsf{a}_t$. Fortunately, this alternation would only be necessary until the estimates of error rates stabilized, which can happen rapidly when we pool across the (assumed) independent sets of input connections. That is, for a population of $N$ neurons, one observes $N$ outcomes for each stimulation $t$, suggesting accurate estimation in only a small number of trials $T$ (provided the $\mathbb{E}\mathsf{a}_t$ estimates are not changing rapidly). We leave this possibility for future work.

## D   Additional experiments

All experimental simulations were run on a 2018 custom-built desktop machine with 128 GB of system memory, a 14 core 3.1 GHz Intel i9-7940X processor, an NVIDIA Titan Xp GPU with 12 GB of memory, and running Ubuntu 18.04.4 LTS.

Figure S4 shows the variation due to setting different random seeds, along with the time per iteration. Each set of results (specificity and sensitivity for all tests) takes about 20 minutes in total to run when using 50 iterations for batch fitting. To run 500 tests for a N=1000 system, for example, would only take up to 3.5 minutes (see specific timing information for each set of tests in Fig. S4c).

Our model has only a few relevant hyperparameters: $\mu$ and $\sigma$ when using the weak entropy bound, as well as the step size for gradient descent or Adam. For best recovery, as defined by higher specificity and sensitivity in the fewest number of tests, we set $\mu = 0$ and $\sigma = 0.1$. In batch mode with Adam, the step size is 0.01, whereas in the streaming case using simple gradient descent, we used a step size of 0.1. Unless otherwise stated, all additional experiments were run using the batch method with base case parameters (N=1000, $\alpha, \beta = 0.05$, S=10, K=$N^{0.3}$).

### D.1   Inference with binary entropy

In contrast with our best recovery approximation, inference with $\mathcal{H}_2(x) = -x\log x - (1-x)\log(1-x)$ (see (14) in main text) requires a much greater number of tests to reach the same level of specificity and sensitivity given a classification boundary of 0.5. Here, we present results for a smaller system, $N = 200$ (Fig. S5). In general, this model exhibits many fewer false positives (specificity $\sim 1$), while the posteriors for the true positive connections are less confident than the approximate case, ranging from 0.5 to 0.8 (when the approximate estimates are $> 0.8$). That is, overconfidence generally benefits recovery performance, while a decision rule based on the posterior marginals from tighter bounds requires many more tests for the same level of accuracy. This is at least in part due to the fact that the marginals fail to capture interactions among the $\mathsf{w}$, and so are expected to underperform estimates like the true MAP, which do.

Figure S4: **Variability and timing. (a, b)** Specificity and sensitivity, respectively, for the base case run with different random seeds (n=20, CI=95%) **(c)** Time per iteration in seconds, averaged across 50 iterations, as a function of the number of tests for the base case.

Figure S5: **Recovery using binary entropy bounds. (a, b)** Specificity and sensitivity, respectively, for different test error rates. **(c)** Calibration plot comparing the weights obtained using the quadratic entropy bound and those obtained with the binary entropy bound (N=200, T=1000, $\alpha, \beta = 0.02$).

## D.2 Sparsity

We additionally tested our method on networks with denser sets of connections, $K = N^\theta$ where $\theta = [0.3, 0.4, 0.5]$, and show that this method is robust to the number of connections per neuron. As the network becomes less sparse, the specificity and sensitivity decrease, but only slightly (Fig. S6).

Figure S6: **Sparsity. (a, b)** Specificity and sensitivity, respectively, for different levels of network sparsity.

Figure S7: **Number of neurons stimulated per test. (a, b)** Specificity and sensitivity, respectively, for different sizes of stimulation groups.

Figure S8: **Recovery in the online and adaptive settings.** Sensitivity corresponding to the specificity plots in the main text Figure 3, as well as online Bernoulli case with a larger window (100 tests).

### D.3 Stimulation group size

In the base case, we used S=10 as the size of our stimulation group. Here we show the effect of varying the stimulation group size (Fig. S7). As S grows, the number of tests required to reach a certain specificity and sensitivity decreases. Indeed, the optimal choice for S is $\frac{1}{K}$ [2] when $K$ is known. However, for $K = N^{0.3} \approx 8$, this number is large (S=125), and larger S (S>20) requires many more iterations of Adam to converge as well as a smaller learning rate (e.g. 200-400 iterations and step size of $\sim 0.001$), making it impractical. Experimentally, it may also make sense to limit the stimulation group size to avoid heating due to repeated photostimulation across large brain areas.

### D.4 Sensitivity plots for online setting

In the streaming setting with 1 iteration per test, sensitivity can drop significantly (more false negatives; see Fig. S8). This is due to the fact that, unlike the batch case, each $a_t$ is only updated for a small number of gradient steps (effectively the window length) before being frozen. The range plotted here, matches the figure in the main text. Performance does eventually also plateau after more tests, and the drop in sensitivity can be ameliorated, by increasing the window size (e.g., from 10 to 100 tests (dotted line)). Ultimately, adaptive stimulation is the most performant: sensitivity is higher overall and plateaus early as a function of the number of tests, since we target the most uncertain connections closest to the 0.5 classifier boundary. As there is a negligible time penalty incurred by using the adaptive method ($\sim$ 1 ms per test for a window length of 10), we suggest only using the adaptive method when fitting online.