[Reviews · NeurIPS 2020]

Review 1

Summary and Contributions: This study develops an approach for inferring whether neurons are functionally connected (binary connection weights), from the observation of the stimulation of small neural ensembles.

Strengths: Soundness of the claims: With one exception (see Weaknesses), the claims are appropriately backed by empirical evaluation. Significance: The results suggest that the developed approach is a solid step towards applications in real neuroscience datasets, given the quality of the inference and the scalability in terms of numbers of inferred connections. Novelty: The novelty of the study is two fold: (1) it uses the framework of Group Testing for network inference; (2) it frames Group Testing as a Bayesian Inference problem. Relevance to the NeurIPS community: The methodology and results presented are of interest to the neuroscience community as well as the Bayesian Inference community and will inspire applications to real neuroscience data, and further method development.

Weaknesses: Soundness of the claims: The discussion about the relation between the proposed approach and previous network inference approaches (such as Aitchison et al. 2017) does not make full justice to these. E.g. the proposed approach is said to have multiple advantages compared to previous approaches (fewer statistical assumptions, ...), but without a proper quantitative comparison with previous methods, such claims should be toned down. Indeed, it could be that approaches relying for instance on generative models have better performance in particular real data settings. So, unless the authors compare their approach with these other methods in terms of specificity, sensitivity and scalability, I would recommend adding a sentence or two safeguarding the possibility that other approaches might be better in certain contexts.

Correctness: The claims and methodology are correct.

Clarity: The paper is clearly written, and provides enough information for an expert reader to understand all the steps to reproduce the results.

Relation to Prior Work: The manuscript clearly puts its contribution in the context of previous work in the field.

Reproducibility: Yes

Additional Feedback: -- After the rebuttal -- The authors' response addressed my comments to satisfaction, and I therefore maintain my support for the (clear) acceptance of this paper.


Review 2

Summary and Contributions: This paper presents an approach to the problem of inferring a functional network across many neurons using noisy group testing. The authors formulate the connections across a population of neurons as a binary network that encodes the presence or absence of functional (not necessarily synaptic) connections between pairs of neurons, with a noisy Bernoulli observation model to capture neurons occasionally not being activated even when neurons they are functionally connected to are stimulated. Inference over the connections is initially formulated as a maximum likelihood problem, which can be rewritten as an integer optimization problem. The authors further extend this formulation by relaxing the variables to restricted continuous values and reformulating the problem as approximate Bayesian inference to infer the posterior probability of each connection. A dual decomposition algorithm is presented for solving the problem, which can be adapted to perform online inference in the setting where an experimenter might want to update the posterior probabilities as new tests are performed or adaptively select tests based on the current network estimate. Experiments on simulated data show the ability of this approach to correctly infer connections as more tests are performed for a range of assumed values of the false positive and negative rates. They also perform experiments that demonstrates the comparable performance of both online and batch inference approaches.

Strengths: The derivation of the problem formulation and algorithm for inferring posterior connection probabilities appears to be sound. In particular, the Bayesian interpretation/formulation that allows the estimation of uncertainty about network connections is an interesting extension and very useful in practice. The empirical evaluation is strong, with various versions of the method tested on a range of different simulated data scenarios, although with no application to real data (more on this below). The contribution seems novel and the authors clearly delineate the differences between theirs and other approaches to the same problem.

Weaknesses: As mentioned above, the primary limitation is the lack of experiments on real data. Particularly since the authors claim that the method is scalable to large datasets, a demonstration of this on a real dataset that shows plausible estimation of a network with a reasonable amount of computation would really strengthen the work.

Correctness: The method and empirical evaluation appear to be correct.

Clarity: Yes, the paper is overall well written.

Relation to Prior Work: Yes, the authors cite several functional connectivity estimation approaches in their introduction, including other group testing approaches like the one they present. In Section 2, they discuss other optimization methods for solving the exact problem they present and how their approach differs from past work. The comparisons to previous work are elaborated upon in the Discussion, with the highlighted differences of the presented approach being the lack of a specific parametric model, the ability to perform either exact or Bayesian inference with a probabilistic interpretation of the inferred parameters, and scalability to large numbers of neurons.

Reproducibility: Yes

Additional Feedback: UPDATE: After reviewing the authors' response, I believe my concerns have been adequately addressed. While I still think that the lack of real data experiments is the main weakness of the paper (and the authors seem to agree), I believe that the well-explained derivation of their approach as well as the extensive empirical evaluations on simulated data provide enough of a technical contribution to warrant acceptance. I have updated my score accordingly.


Review 3

Summary and Contributions: Holographic photostimulation techniques have advanced to the stage it is possible to precisely stimulate selected ensembles of neurons. This paper asks how, mathematically, one should use this new experimental ability to efficiently infer functional connectivity between neurons. The algorithmic approach that is developed is based on noisy group testing and the result shows the number of tests (stimulations) grows logarithmically with population size. This is further extended to a streaming setting. Simulations corroborate theoretical findings. Note that the focus is on the decoding algorithm (which is connected to Bayesian inference) rather than the pooling scheme, which is basically a random design. After author response, I realize I missed their use of adaptive pooling designs, which improves my assessment of the paper.

Strengths: * This problem area is certainly relevant to the neuroscience wing of NeurIPS and has seen many previous papers * The mathematical connection to group testing seems novel: previous work in this area seems to have focused on connections to compressed sensing * Implementation of recovery algorithm via GPUs is interesting for a massive pooling setting as here

Weaknesses: * No experiments with real neuronal data. * No consideration of better pooling designs than random pooling * No consideration of inferring connection weights

Correctness: Paper is correct, as far as I can tell.

Clarity: Paper reads well.

Relation to Prior Work: There seems to be no discussion of compressed sensing approaches to a very closely related problem of multineuron stimulation, which is a somewhat glaring omission. See e.g. the papers [Hu, Leonardo, Chklovskii, "Reconstruction of sparse circuits using multi-neuronal excitation (RESCUME)", 2009] and [Fletcher, et al, "Neural reconstruction with approximate message passing (NeuRAMP)", 2011].

Reproducibility: Yes

Additional Feedback:


Review 4

Summary and Contributions: This paper proposes a novel approach for inferring the functional connectivity in a network of neurons, which involves making an intervention (stimulation) of a selected group of neurons in the network, and testing which neurons changed state in response to the stimulation. The paper claims that the method scales well, primarily due to the group-based nature, especially in the sparse regime. A notable contribution of this work is how it formulates the inference problem for the connectivity: it is first written down in terms of the likelihood of group test results, then as a constrained optimization problem with relaxed variables, as a Bayesian inference problem, then again as an optimization problem for probability distribution in a variational Bayes. The optimization problem is analyzed in a rigorous statistical light. Finally, a simulated test demonstrates that the proposed method can recover a sparse network in a readily scalable way.

Strengths: This paper provides a fresh perspective to the important problem of functional connectivity inference. The subject is relevant to the broad neuroscience community, and also timely, given the increasing access to large-scale neural recordings. This paper will be of great interest to the NeurIPS community. The contribution is clearly novel, and its impact could be significant if (i) the proposed method can be implemented experimentally, with photostimulation or other forms of intervention, and (ii) an easy-to-use software package can be developed. The theoretical development of the inference method is very impressive; I feel that the paper could merit acceptance just for this, even without taking into account the potential practical benefit.

Weaknesses: The practical strength of the method should be tested through actual experimental implementations, which would be a future work. The proposed method requires certain conditions (statistical assumptions), and it may take a large number of stimulations, as discussed in the paper.

Correctness: The formulation of the group testing and the inference problems appear to be correct. I do not have the full expertise to assess the correctness of the optimization theory and the statistics involved, but the construction seems to be grounded on a well-established body of previous works.

Clarity: This paper is very well written; it is with an exceptional clarity that the method sections unfold, given all the technical complexity. The paper uses an effective story-telling style, with a series of goal-challenge-solution blocks. It also provides many insightful comments that helps appreciating the work in a broader context.

Relation to Prior Work: The paper does a great job of putting its contributions in context with the previous works. Such contexts are provided throughout the paper almost every time a modeling choice is made (by mentioning the alternatives); several paragraph in the discussion section is then dedicated to discussing prior works in the two related directions, group testing and functional connectivity inference.

Reproducibility: Yes

Additional Feedback: Great work --- it was a stimulating read. I wonder if the authors are thinking about an experimental implementation of this method; can you comment if you are working on a follow-up? And what would be the potential problems if this method was to be made more broadly available in many neurophysiology labs, and for different brain areas? ======= ** Edit ** The authors response addressed my concern about the experimental validation of the method, the major limitation of this paper as agreed among the reviewers. I remain enthusiastic about this work despite the limitation, and I also look forward to seeing follow-ups in the future.

[Author Response · NeurIPS 2020]

We thank the reviewers for their constructive comments on our submission. Below we address the raised concerns and include clarifications as suggested.

**Experimental data.** All reviewers noted a lack of experiments conducted on real data as a weakness of the work. We agree. As R4 suggests, we are ourselves working to validate these ideas experimentally. But there are two serious difficulties with using existing data in our context: **First,** because ground truth connectivity is almost never known for real data *in vivo*, most previous work, like ours, evaluates correctness on synthetic data, including (1; 2). **Second,** since we are modeling responses to stimulation of specific neurons, we require a dataset comprising similar manipulations. We know of no good benchmark data for these purposes, though we aim to generate them in future. Thus, our immediate follow-up plans involve performing simulations in more biologically plausible spiking networks, including nonlinear effects of photostimulation that more closely match biological responses (3; 4). This will allow us to directly compare our model to parametric approaches like (2; 5), since we agree with R1 that there are indeed likely to be cases and datasets in which these models outperform ours.

**Relation to compressed sensing.** We thank R3 for pointing out the highly relevant paper (1) and apologize for the omission. We now include this work (along with (2) in our discussion of inference from observation data using parametric models). While the manuscript already discusses relationships between our approach and one-bit compressed sensing, we had missed this work. **Key differences** between the present work and (1) include: **(a)** For speed and scalability, we focus on recovering binarized (present/absent) connections, not full weights. In practice, this may be all experiments require, and when it does not suffice, our approach may be used to rapidly pre-screen connections before performing more focused testing (an approach also suggested in (1)). **(b)** While the CoSaMP approach of (1) assumes a *known* level of sparsity, our method does not. In fact, our incorporation of uncertainty in weights allows for optional stopping. **(c)** We demonstrate scaling and speed necessary for implementation in the online setting. We plan to more fully discuss all these issues in our revised manuscript.

**Theoretical note.** In comparison with compressed sensing, we note an interesting connection: while our $a_t = \|\mathsf{w} \odot \mathsf{x}_t\|_\infty$, the equivalent linear predictor for 1-bit CS is $\|\mathbf{w} \odot \mathsf{x}_t\|_1$, and our relaxed $a_t$ is constrained to lie between these two. This raises the possibility that a generalization of our model might be able to interpolate between the two approaches.

**Stimulation types.** R3 noted as a weakness that we only consider randomized stimulation groups. This is incorrect. In our experiments, we also consider selecting subgroups adaptively by choosing to stimulate neurons with maximum marginal uncertainty (cf. Figure 3 in main text). As we show, this results in performance improvements over fully randomized stimulation.

**Our contributions.** While there has been much previous work on inferring network connectivity, as all reviewers note, the present work also contains novel methodological advances of broader interest, including: **(a)** the first application of group testing to network inference in neuroscience; **(b)** a novel relaxation of group testing, along with an equivalence to variational Bayesian inference; **(c)** a fast dual decomposition algorithm and GPU software implementation that makes online inference practical in large networks.

# References

[1] Hu, T., A. Leonardo, and D. B. Chklovskii. "Reconstruction of sparse circuits using multi-neuronal excitation (RESCUME)". In Advances in Neural Information Processing Systems 22, pp. 790-798 (2009).

[2] Fletcher, A. K., S. Rangan, L. R. Varshney, and A. Bhargava. "Neural reconstruction with approximate message passing (NeuRAMP)". In Advances in Neural Information Processing Systems 24, pp. 2555-2563 (2011).

[3] Charles, A. S., A. Song, J. L. Gauthier, J. W. Pillow, and D. W. Tank. "Neural Anatomy and Optical Microscopy (NAOMi) Simulation for evaluating calcium imaging methods". bioRxiv:10.1101/726174 (2020).

[4] Luboeinski, J., and T. Tchumatchenko. "Nonlinear response characteristics of neural networks and single neurons undergoing optogenetic excitation". Network Neuroscience Advance publication. doi.org/10.1162/netn_a_00154 (2020).

[5] Aitchison, L., L. Russell, A. M. Packer, J. Yan, P. Castonguay, M. Hausser, and S. C. Turaga. "Model-based Bayesian inference of neural activity and connectivity from all-optical interrogation of a neural circuit". In Advances in Neural Information Processing Systems 30, pp. 3486-3495 (2017).


[Meta-Review · NeurIPS 2020]

Nice work. The group testing methods and connections to VI for neural connectivity inference will be of great interest to the NeurIPS community. In addition to the incorporating the reviewers' feedback into the final paper, please also address these minor issues: - You simply assert the marginal polytope constraints in eq (7). Please explain how these are derived from the exponential family form of Q. - Please also cite this relevant past work at NeurIPS: Shababo B, Paige B, Pakman A, Paninski L. Bayesian inference and online experimental design for mapping neural microcircuits. In Advances in Neural Information Processing Systems 2013 (pp. 1304-1312). - Shababo and Paninski (together with Shizhe Chen and Hillel Adesnik) have been extending and applying these methods, but I don't know if it has been published.